# Which Three-Dimensional Printing Technology Can Replace Conventional Manual Method of Manufacturing Oral Appliance? A Preliminary Comparative Study of Physical and Mechanical Properties

**Hyo-Jin Kim** [1,2,†], **Seung-Weon Lim** [2,3,4,†], **Mi-Kyung Lee** [2], **Sung Won Ju** [5], **Suk-Hee Park** [6], **Jin-Soo Ahn** [5] **and Kyung-Gyun Hwang** [2,4,7,*]

1   Department of Medical and Digital Engineering, Hanyang University, Seoul 04763, Korea; khj1017@hanyang.ac.kr
2   Smart Oral Health Care Research Center, Hanyang University, Seoul 04763, Korea; swlim@hanyang.ac.kr (S.-W.L.); vision21lmk@gmail.com (M.-K.L.)
3   Division of Orthodontics, Department of Dentistry, Hanyang University Medical Center, Seoul 04763, Korea
4   Department of Dentistry, College of Medicine, Hanyang University, Seoul 04763, Korea
5   Department of Dental Biomaterials Science, School of Dentistry and Dental Research Institute, Seoul National University, Seoul 03080, Korea; softwareju@gmail.com (S.W.J.); ahnjin@snu.ac.kr (J.-S.A.)
6   School of Mechanical Engineering, Pusan National University, Busan 46241, Korea; selome815@pusan.ac.kr
7   Division of Oral & Maxillofacial Surgery, Department of Dentistry, Hanyang University Medical Center, Seoul 04763, Korea
*   Correspondence: hkg@hanyang.ac.kr; Tel./Fax: +82-2-2290-8676
†   These authors contributed equally to this work.

**Abstract:** Three-dimensional printing technology is widely being adopted in the manufacturing of oral appliances. The purpose of this study was to determine the most suitable method of manufacturing oral appliances by comparing the physical and mechanical properties of various 3D printing methods with the conventional method. Experimental groups consisted of six 3D-printed specimens via FDM, two polyjets, SLS, SLA, and DLP, and the milling methods. The control group consisted of an acrylic resin specimen made by the conventional manual method. The water absorption and solubility, color stability, flexural strength, and surface hardness were tested and statistically analyzed. The FDM, SLS, and DLP methods exhibited comparable water absorption and solubility with the control group, and only the SLA method exhibited significantly higher water solubility than the control group. In terms of the color stability, only the milling method met the requirements of the allowable clinical range. The FDM, SLA, and DLP methods exhibited comparable flexural strength with the control group. The surface hardness of the PJ-2, DLP, and milling methods was acceptable for replacing conventional manual method. Therefore, the most suitable method of manufacturing oral appliances among the experimental groups was the DLP method in terms of its water absorption and solubility, flexural strength, and surface hardness.

**Keywords:** 3D printing technology; water absorption; water solubility; color stability; flexural strength; surface hardness

## 1. Introduction

Oral appliances, which are used to manage oral and maxillofacial dysfunctions, are made to cover the teeth and mucous membrane of the oral cavity, of which we list some examples. The orthodontic clear aligner is used to control tooth movement [1–4]. The implant surgical guide is used to place an implant into an appropriate position [5]. The mandible advancement device (MAD) is used to widen the posterior airway space by changing the position of the jaw and tongue [6–8]. Occlusal splints are used to manage disc displacement of the temporomandibular joint [9–11]. Although many oral appliances are widely used for

their effectiveness, compromised accuracy and reliability due to the conventional multi-step laboratory procedure often make them difficult to apply in everyday clinics.

Various industries have already been using additive manufacturing profitably for many years. Many additive manufacturing produced parts are already incorporated into aerospace engines and vehicles [12]. Medical companies value the ability of additive manufacturing to convert patient-specific data for customized products and medical interventions [13,14]. Accordingly, digital technology was introduced in dentistry, and new changes have been made in the manufacturing methods of oral appliances, which can largely be divided into subtractive milling methods and additive 3D printing methods. A computerized milling system is involved with the procedures of selective reduction of pre-manufactured blocks to the final shape with a cutting tool. On the contrary, three-dimensional (3D) printing technology implements laminar manufacturing of the final structure; hence, not only this method is free of restricted movement of cutting tools, the prices can also be more economic than the subtractive milling method [15–18]. Therefore, the application of 3D printing technology has rapidly increased recently in dentistry [18].

The method of 3D printing can be divided into various methods according to the technology and material used. The fused deposition modeling (FDM) method uses extruded thermoplastic materials and was commercialized in the industry [19]. Powder-based 3D printers such as selective laser sintering (SLS) use nylon or a similar thermoplastic powder that is locally melted with a laser beam. Recently, liquid-based 3D printing technologies such as stereolithography apparatus (SLA) and digital light processing (DLP) and polyjet (PJ) have been adopted. Ultraviolet-curable resin is polymerized to form the desired shape by light sources in these technologies.

These various 3D printing technologies would be applied in oral appliance manufacturing in the near future. Prototypes of the occlusal splint fabricated with various fabrication methods are shown as Figure 1. However, several important physical and mechanical properties need to be evaluated for the clinical application of 3D-printed devices. Specifically, 3D-printed appliances should be safe in the oral cavity, which is an environment with dynamic chemical, physical, and biological changes, and meet the criterion of the ISO (International Standardization for Organization). If unqualified properties of the 3D-printed oral appliances are found, endeavors to improve these inferior properties should be undertaken. Until now, many clinicians have used the familiar, conventional manual method for manufacturing the oral appliances. Therefore, the properties such as water absorption and solubility, color stability, flexural strength, and surface hardness of the 3D-printed products manufactured with various 3D printing methods need to be compared with the conventional manual method and with each other. The objective of this study was to determine the most suiTable 3D printing method for manufacturing oral appliances by comparing the physical and mechanical properties of the various 3D printing methods with the conventional manual method. The null hypothesis of this study was that there would be no differences in the physical and mechanical properties of the 3D printing technologies when compared with the conventional manual method.

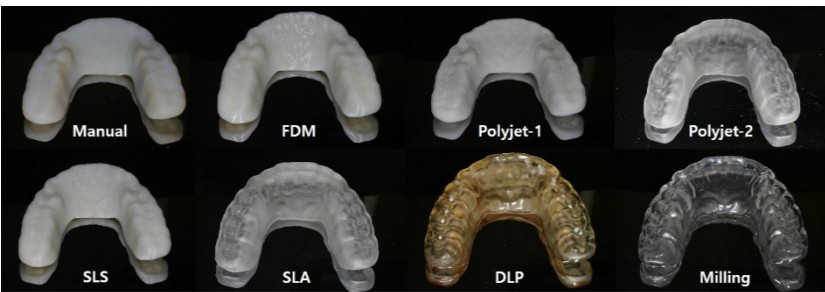

**Figure 1.** Occlusal splint fabricated with manual method, various 3D printing technologies, and the milling method.

## 2. Materials and Methods

### 2.1. Preparing the Test Specimen

Table 1 summarizes the composition of the 7 experimental groups and control group in this study. Six of the main experimental groups consisted of specimens 3D-printed via FDM, 2 PJs (PJ-1 and PJ-2), SLS, SLA, and DLP. The last experimental group consisted of subtractive-manufactured specimens milled from a clear polymethyl methacrylate (PMMA) block. The control group consisted of acrylic resin specimens made by the conventional manual method. A total of 10 specimens were prepared for each group.

**Table 1.** The composition of the 7 experimental groups and control group in this study.

| Method | Material | Equipment | Layer Thickness (mm) |
|---|---|---|---|
| Manual (control) | Acrylic resin | Ortho jet (Lang Dental Manufacturing Company, Inc., Wheeling, IL, USA | - |
| FDM | ABS-M30 | FORTUS 450MC (Stratasys Inc., Hennepin County, MN, USA) | 0.1 |
| PJ-1 | Vero-clear | J750 (Stratasys Inc., Hennepin County, MN, USA) | 0.03 |
| PJ-2 | Med-610 | OBJET260 CONNEX 2 (Stratasys Inc., Hennepin County, MN, USA) | 0.05 |
| SLS | Polyamide | sPRO 140 (3Dsystems, Wilsonville, OR, USA) | 0.06 |
| SLA | Exclusive Form2 clear resin | Form2 (Formlabs, Middlesex County, MA, USA) | 0.05 |
| DLP | NextDent ortho Rigid | NextDent 5100 (3Dsystems, Wilsonville, OR, USA) | 0.025 |
| Milling | Clear PMMA block | ARUM 5X-200 (Arum dentistry Co., Ltd., Daejeon, Korea) | - |

FDM, fusion deposition modeling; ABS, acrylonitrile butadiene styrene; PJ, polyjet; SLS, selective laser sintering; SLA, stereolithography apparatus; DLP, digital light processing; PMMA, polymethyl methacrylate. Manual method is the control group, and the other methods are experimental groups.

### 2.2. Experiments

2.2.1. Water Absorption and Solubility

The specimens were prepared by exporting the disc of stereolithography (STL) format into each 3D printer and the milling system. A total of 10 specimens per group were cut using a specimen cutter and polished using a surface grinder with 2000 grit sandpaper to produce specimens (50 mm diameter, $0.5 \pm 0.1$ mm thickness). In this study, storage of specimens in silica gel was performed for the absorption and solubility tests. After drying the 8 types of specimens for $23 \pm 1$ h in a dryer ($37 \pm 1$ °C) containing silica gel, the cycle was repeated for $60 \pm 10$ min in the second dryer maintained at $23 \pm 2$ °C. The specimens were taken out one by one, and the weight was measured with an accuracy of 0.1 mg ($m_1$) (XS204 Delta range, Mettler Toledo, Columbus, OH, USA). The volumes (v) of the specimens were calculated using the mean diameter and mean thickness. Each specimen was immersed in 20 mL of distilled water ($37 \pm 1$ °C) for 7 days $\pm$ 2 h; then each specimen was taken out, the moisture was completely removed, and the weights ($m_2$) of the specimens were recorded. Each specimen was reprocessed in the dryer until it reached a constant weight, and the final weight was measured and recorded ($m_3$). The water absorption ($W_1$) and water solubility ($W_2$) of each specimen was measured in $\mu g/mm^3$ using the following equation.

$$\text{Water absorption } (W_1) = \frac{m_2 - m_3}{V} \tag{1}$$

$$\text{Water solubility } (W_2) = \frac{m_1 - m_3}{V} \tag{2}$$

2.2.2. Color Stability

The specimens were constructed as square blocks with widths and lengths of 10 mm and thicknesses of 3 mm using the Rhino program. Each group consisted of 3 specimens. The specimens were immersed in erythrosine 3% at 37 °C. Color change ($\Delta E$) was measured

by a spectrophotometer (Color-Eye 7000A, Greta Macbeth, München, Germany) before immersion, and at 10, 20, and 30 days after immersion. The color Commission Internationale d'Eclairage (CIE) Lab values were measured at 3 random areas. The CIELAB tri-stimulus X, Y, and Z values were shown as CIE $\Delta$L (brightness white-black), $\Delta$a (red-green), and $\Delta$b (yellow-blue).

Color change ($\Delta$E) values were calculated from the following expressions.

$$\Delta E = \sqrt{\Delta L^2 + \Delta a^2 + \Delta b^2} \tag{3}$$

$$\begin{aligned} \Delta L = L2 - L1 \ (L2 = \text{value after immersion, } L1 = \text{value before immersion (day0)}) \\ \Delta a = a2 - a1 \ (a2 = \text{value after immersion, } a1 = \text{value before immersion (day0)}) \\ \Delta b = b2 - b1 \ (b2 = \text{value after immersion, } b1 = \text{value before immersion (day0)}) \end{aligned} \tag{4}$$

### 2.2.3. Flexural Strength

In accordance with the international standard of ISO 20795-2, the specimens were produced as a rectangular plate with widths of 64 mm, lengths of 10 mm, and thicknesses of 3 mm. Each group consisted of 10 specimens. The flexural strength of each specimen was tested using a universal testing machine with a 1.0 mm/min loading rate (Instron, Norwood, MA, USA) (Figure 2). The load was applied at the middle of the specimens until the specimen was fractured. A maximum load of 800 kg was applied. The maximum load (N) applied to the specimen was measured and the flexural strength (MPa) was calculated by the equation below.

$$\text{Flexural strength F} = \frac{3P1L}{2BH^2} (MPa) \tag{5}$$

$P_1$: Force at the time of the destruction of the specimen (N)
L: Spacing between supports
B: Width of the specimen measured prior to the test
H: The thickness of the specimen measured prior to the test

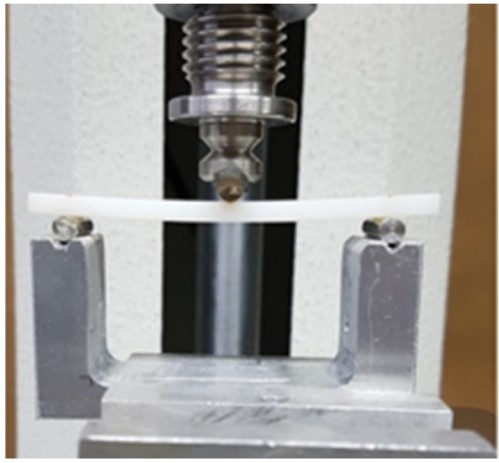

**Figure 2.** Flexural strength test.

### 2.2.4. Surface Hardness

In accordance with the international standard of ISO 20795-2, specimens were produced the same as the flexural strength test. To reproduce an oral environment, each of the specimens was immersed in distilled water at 37 °C $\pm$ 1 °C for 24 h. Afterwards, the surface hardness was measured 3 times on each specimen to obtain the average value using a Vickers hardness machine (DMH-2, Matsuzawa Siki Co., Ltd., Akita, Japan).

### 2.2.5. Fractured Surface Observation

A scanning electron microscope (SEM) was used to observe defects and the microstructures of the specimens at a nano-level. The fractured surfaces after testing the flexural strength and non-fractured surfaces in case of the non-fracture specimen were observed. In the manual, FDM, PJ-1, PJ-2, SLA, DLP, and milling specimens, shredded wave surfaces were observed at 30×, 500×, and 10,000×, while unbroken curved surfaces were observed in the SLS specimen.

### 2.3. Statistics Processing

To compare water absorption and solubility, flexural strength, and surface hardness among the 8 types of specimens, data were analyzed using the Kruskal–Wallis test followed by the Mann–Whitney U test using the Bonferroni correction, where the level of significance was adjusted accordingly. A *p*-value of <0.05 was considered statistically significant. All the statistical analyses were performed using the SPSS software (version 24.0; IBM, Armonk, NY, USA).

## 3. Results

### 3.1. Experiments

### 3.1.1. Water Absorption and Solubility

Figure 3 and Table 2 show significant differences in water absorption between the groups. The post-hoc analysis demonstrated that significance was attributed to differences between the conventional manual with two polyjets, SLA, and milling methods. The water absorption of SLA, PJ-1, PJ-2, and milling were 52.31, 35.00, 31.60, and 30.69 µg/mm$^3$, respectively, while that of manual was 18.51 µg/mm$^3$. There was no significant difference between the two polyjet groups. The water absorption, in order of highest to lowest, was SLA, PJ-1, PJ-2, milling, FDM, SLS, manual, and DLP. Only the SLA group, which showed the highest absorption rate of 52 µg/mm$^3$, did not meet the requirements of ISO 10477 (50 µg/mm$^3$ or less). The DLP group demonstrated the lowest water absorption rate of 17 µg/mm$^3$.

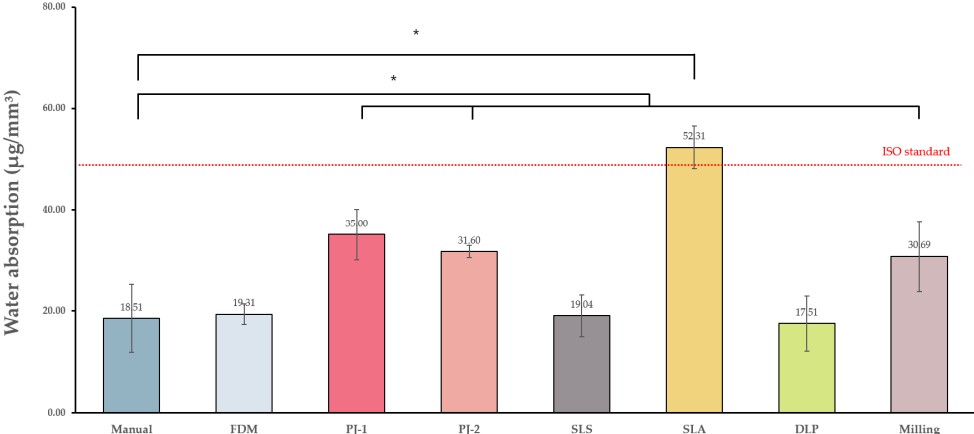

**Figure 3.** Comparison of water absorption of each group. * Denotes *p* < 0.05/28 for comparison between the manual and experimental groups; when 50 µg/mm$^3$ or less, ISO10477 is satisfied (based on the red line).

**Table 2.** Value of the water absorption, solubility, color stability, flexural strength, and surface hardness of the eight tested groups.

| Classification | Group | Absorption (μg/mm³) Mean ± SD | Solubility (μg/mm³) Mean ± SD | Color Change (ΔE) Mean ± SD | Flexural Strength (Mpa) Mean ± SD | Surface Hardness (HB) Mean ± SD |
|---|---|---|---|---|---|---|
| Control group | Manual | 18.51 ± 6.69 [a] | 5.95 ± 2.84 [ab] | 13.34 ± 16.12 [a] | 62.12 ± 5.99 [a] | 83.37 ± 1.5 [a] |
| Experimental groups | FDM | 19.31 ± 2.02 [a] | −1.43 ± 0.49 [a] | 32.58 ± 0.94 [b] | 58.84 ± 0.39 [a] | 75.08 ± 2.33 [b] |
| | PJ-1 | 35 ± 5.05 [b] | −1.06 ± 1.19 [a] | 31.82 ± 1.67 [b] | 83.71 ± 1.24 [b] | 77.82 ± 0.86 [b] |
| | PJ-2 | 31.6 ± 1.2 [b] | −1.55 ± 0.38 [a] | 24.67 ± 2.96 [b] | 94.95 ± 2.94 [b] | 83.12 ± 0.82 [a] |
| | SLS | 19.04 ± 4.12 [a] | −0.63 ± 0.87 [a] | 60.95 ± 4.94 [c] | 41.15 ± 2.58 [a] | 73.8 ± 5.58 [b] |
| | SLA | 52.31 ± 4.17 [c] | 15.6 ± 6.29 [c] | 12.76 ± 1.97 [a] | 69.37 ± 2.1 [a] | 79.62 ± 0.63 [b] |
| | DLP | 17.51 ± 5.39 [a] | 5.42 ± 3.64 [b] | 14.26 ± 4.61 [a] | 72.96 ± 17.63 [a] | 86 ± 0.58 [c] |
| | Milling | 30.69 ± 6.9 [b] | 5.11 ± 4.25 [b] | 2.82 ± 0.29 [a] | 120.36 ± 6.63 [c] | 87.68 ± 1.21 [c] |

SD, standard deviation; FDM, fusion deposition modeling; PJ, polyjet; SLS, selective laser sintering; SLA, stereolithography apparatus; DLP, digital light processing; having the same characters (a–c) indicates no statistically significant difference.

Figure 4 and Table 2 show significant differences in water solubility between the groups. The post-hoc analysis demonstrated that significance was attributed to differences of the SLA method with the other methods. The water solubility of the SLA method was 15.60 μg/mm³, while those of other methods ranged from −1.55 to 5.96 μg/mm³. Only the SLA group did not meet the requirements of ISO 10477 (7.5 μg/mm³ or less).

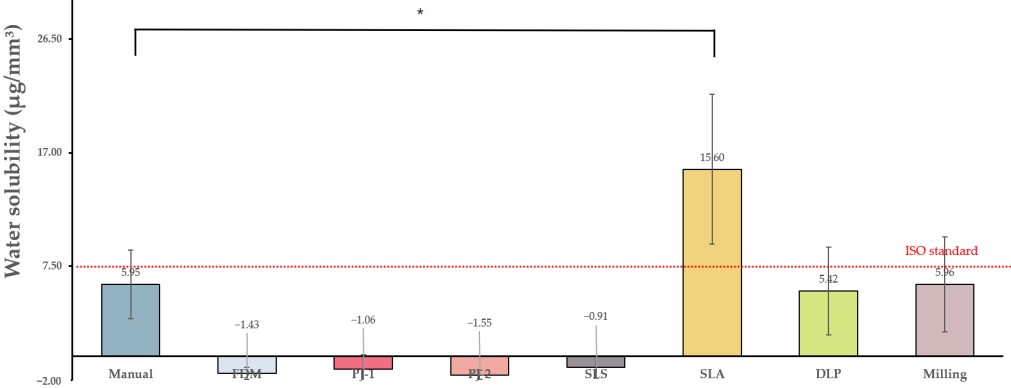

**Figure 4.** Comparison of water solubility of each group. * denotes $p < 0.05/28$ for comparison between the manual and experimental groups; when 7.5 μg/mm³ or less, ISO10477 is satisfied (based on the red line).

The FDM, SLS, and DLP methods exhibited water absorption and solubility comparable with the conventional manual method and acceptable values based on the ISO standard.

### 3.1.2. Color Stability

Figure 5 and Table 2 show significant differences in color stability between the groups. Post-hoc analysis demonstrated that significance was attributed to differences between the manual and FDM, two polyjets, and SLS methods. The color change (ΔE) of the FDM, PJ-1, and PJ-2, and SLS groups was 32.58, 31.82, 24.67, and 60.95, respectively, while that of the manual group was 13.34. The color stability was, in the order of highest to lowest, milling, SLA, manual, DLP, PJ-2, PJ-1, FDM, and SLS. Only milling, which showed the lowest color change of 2.82, met the requirements of the clinical allowable range (ΔE 3.3 or less).

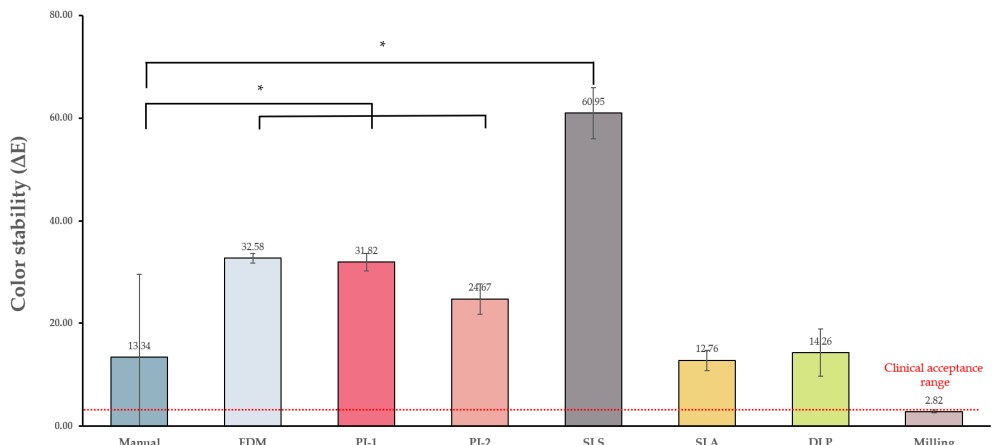

**Figure 5.** Comparison of color stability of each group. * denotes $p < 0.05/28$ for comparison between the manual and experimental groups; when ΔE was 3.3 or less, it satisfies the clinical acceptance range (based on the red line).

### 3.1.3. Flexural Strength

Figure 6 and Table 2 show significant differences in flexural strength between the groups. The post-hoc analysis demonstrated that significance was attributed to differences between the manual and two polyjets and milling methods. The flexural strengths of the PJ-1, PJ-2, and milling methods were 83.71, 94.95, and 120.36 MPa, respectively, while that of the manual group was 62.12 MPa. There was no significant difference between the two polyjet methods. The flexural strength was, in the order of highest to lowest, milling, PJ-2, PJ-1, DLP, SLA, manual, FDM, and SLS. Only the SLS group, which showed the lowest flexural strength of 41.15, did not meet the requirements of ISO 20795-2 (50 MPa or higher). The milling method demonstrated the highest flexural strength of 120.36 MPa. The FDM, SLA, and DLP methods exhibited a flexural strength comparable with the conventional manual method and acceptable values based on the ISO standard.

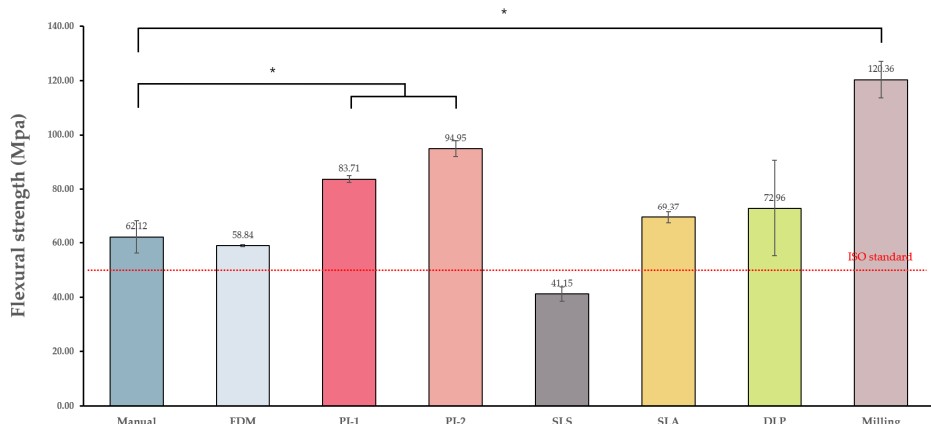

**Figure 6.** The mean and standard deviations of flexural strength. * denotes $p < 0.05/28$ for comparison between the manual and experimental groups; when 50 MPa or more, ISO20795-2 is satisfied (based on the red line).

### 3.1.4. Surface Hardness

Figure 7 and Table 2 show significant differences in surface hardness between the groups. The post-hoc analysis demonstrated that significance was attributed to differences between the manual and all other methods except the PJ-2 method. The surface hardness of DLP and milling were significantly higher than for the manual method, while those of FDM, PJ-1, SLS, and SLA were significantly lower than the manual method. The surface hardness was highest in the order of milling, DLP, manual, PJ-2, SLA, PJ-1, FDM, and SLS.

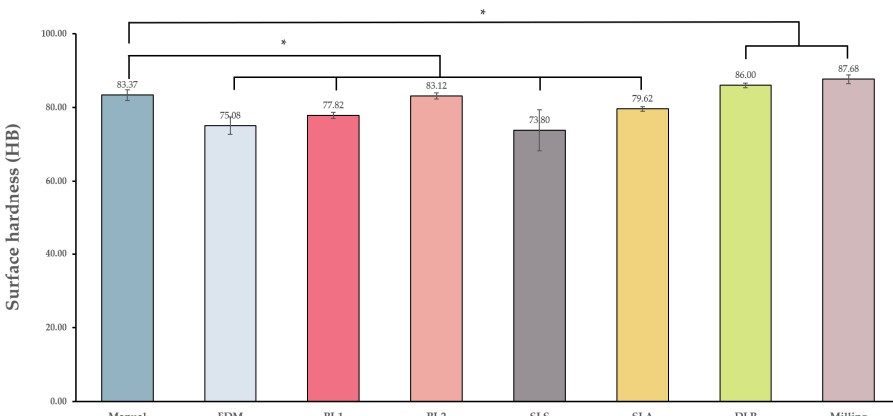

**Figure 7.** Mean and standard deviations of the hardness. * denotes $p < 0.05/28$ for comparison between the manual and experimental groups.

3.1.5. Fractured Surface Observation

Figure 8 shows SEM images of the fractured surfaces of the tested specimens. Small round grains were evenly distributed in the manual group. In the case of FDM, voids generated during the construction of filaments were observed through a magnification of ×30, where each layer was thick, and the molten filaments were thinly stretched in the last layer. The DLP showed a sharper cut surface than manual, and the flow of cured material was not flat. In the cases of PJ-1 and PJ-2, the cut surfaces were smooth, while the regular surfaces were rough and uneven. A smooth surface was presented in the case of the SLS because specimens were not fractured during the flexural strength test. In SLA and DLP, there were signs of vertical cleavage with respect to the fracture direction, which was observed much more clearly in SLA than in DLP. On the other hand, in the milling group, a small number of cracks and torn traces were seen along the uniform fractured surface, and a recessed shape was observed.

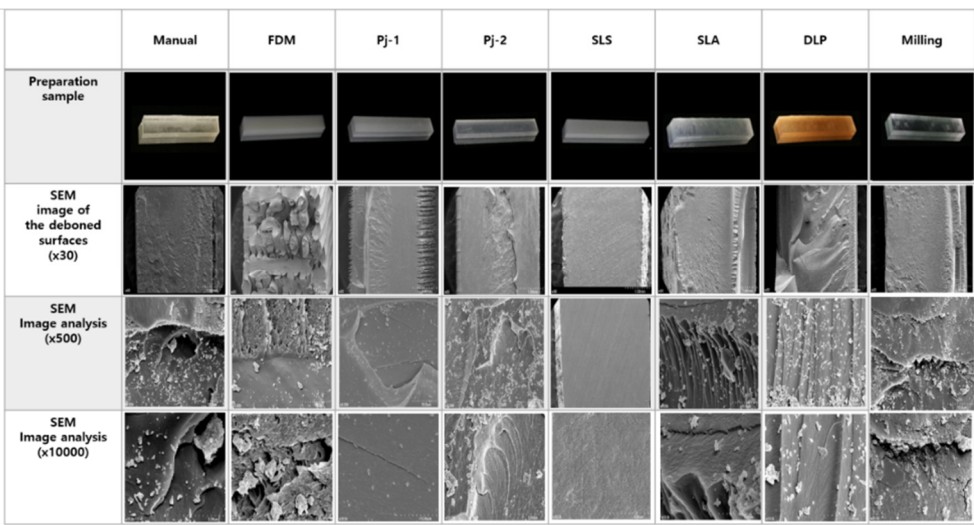

**Figure 8.** Preparation samples and SEM image analysis.

**4. Discussion**

Recently, various 3D printing technologies with different materials have been marketed to clinicians and are being used for oral appliance manufacturing. In this study, we compared widely used six types of 3D printing materials, milling material, and conventional acrylic resin, in terms of the water solubility and absorption, color stability, flexural strength, surface hardness, and observed the fractured surfaces to determine if the 3D-printed materials can replace conventional acrylic resin.

The physical characteristics of polymers are altered by moisture via their plasticizing effects. Indrani et al. have reported that moisture causes increases in plasticity, where expansion of these plastic areas triggers cracks, finally leading to resin degradation [20]. In addition, volume changes in oral appliances under high moisture conditions might affect their suitability and precision. Therefore, water absorption and solubility tests should be conducted. The SLA methods showed significantly higher water absorption and solubility values compared to other 3D-printing and manual methods. This result is consistent with previous studies [21]. The mean absorption of the SLA method was 52.31 µg/mm$^3$, which is about a 2.8-fold increase compared to that of the manual method, which has a value of 18.51 µg/mm$^3$. Therefore, the SLA method is not suitable for replacing the conventional method in manufacturing intraoral appliances. The FDM, SLS, DLP, and milling methods were not only acceptable based on the ISO standard, but also exhibited values directly comparable with the manual method.

It is the case that 3D-printed material can be discolored by internal factors such as the substrate of the resin, as well as external factors such as food consumed. Discoloration often reduces esthetics, which is relevant to patient satisfaction. Therefore, the materials used in oral appliances need to be evaluated for color stability to ensure long-term esthetics. Color stability is quantitatively expressed using ΔE values. If the ΔE value is less than 1, the observer cannot typically recognize any differences, and if the ΔE value is between 1 and 3.3, a skilled technician can easily recognize any differences; however, if the ΔE value is greater than 3.3, it is clinically unacceptable [22]. In all tested methods except for milling, significant color changes were demonstrated, which far exceeded the clinically problematic level of 3.3. This result was consistent with the previous study by Gruber et al., which compared the color changes for manual, milling, and 3D-printed materials. They reported that 3D printing materials showed low color stability, despite there being no significant difference between the manual and milling methods [23]. The possible reason for the low color stability of the 3D-printed materials is that there can exist additional layers in the surface microstructure, since 3D printing is based on an additive manufacturing method [24–26]. Looking at the SEM results, although the surface characteristics of the specimens were different depending on the 3D printing method used, it was possible to observe the microstructure reflected by the pattern structure of the surface (Figure 8). Since the DLP method uses a micro-mirror, a slightly more characteristic pattern appears on the surface, which may cause color stability to decrease. However, the surface roughness for ungrounded milling is higher in the milling specimens than in the 3D printer resin specimens. Therefore, the low color stability of the 3D-printed specimens cannot be explained by the rough surfaces. Instead, the low polymerization rate of the 3D printing resin may be another cause of the lower color stability compared to other materials [26].

Flexural strength is a measurement of resistance to stress by determining the limitations of the material's stress to resist deformation when applying a force with a certain speed to the specimen. It is possible to predict the life expectancy and assess the reliability of materials through their flexural strength. A three-point flexural strength test was conducted in this study, which can evaluate the mechanical properties relevant to withstanding coherence pressures of the oral appliances [27]. In the flexural strength experiments, only the SLS group, which showed the lowest flexural strength of 41.15, did not meet the requirements of ISO 20795-2. This was because the SLS specimen was not broken during the process of measuring bending strength, and only a degree of bending was observed. Therefore, an oral appliance is likely to be deformed by strong force instead of being fractured when manufactured using the SLS method. Similarly, previous studies have shown that polyamide, which we used in SLS method, has many advantages when selecting inelastic materials [28,29]. The DLP, SLA, and FDM methods were acceptable based on the ISO standard and have values comparable with the manual method. In Schruti's study, the flexural strength of the DLP method was 75 MPa, which was equivalent to ours (72.96 MPa) [30]. The two polyjets and milling methods showed significantly higher flexural strengths than the manual method. Regarding the polyjet methods, previous studies also showed flexural

strength values of 78.8 MPa and 95.66 MPa [31], which are similar to the results of the present study. Milling showed the largest difference from manual, being about twice as high as the manual value of 62.12 MPa. Previous studies have also shown that the flexural strength of milling is higher than those of other 3D printer methods [30,31]. However, a high flexural strength may concentrate excessive stress on oral appliances, thus limiting the manufacture of oral devices.

The image of a fractured surface could provide the clinicians with the information of micro-topography which reflects the 3D printing processes (Figure 8). In addition, the voids inside the material could provide an explanation of the resulting flexural strength among the five fractured 3D-printed methods [28]. The highest number of craters and voids was observed in material extrusion printing, the FDM, which could be the reason for the FDM having the lowest flexural strength. The SLA and DLP methods exhibited comparatively similar surfaces with several vertical cleavages; hence, these two methods resulted in not merely similar but higher strength than the FDM method. Lastly, the two PJ methods which exhibited smooth cut surfaces had the highest flexural strength among the five fractured 3D printing methods (Figures 6 and 8).

Surface hardness is a mechanical property that indicates the magnitude of resistance to permanent deformation. A sufficient surface hardness of an oral appliance is needed to endure occlusal and other external forces, which is relevant to the durability of the oral appliance [32]. The results of this study showed that the FDM and SLS methods have lower surface hardness values than the manual method. Previous studies also reported that polyamide materials, which were used in the SLS group, resulted in lower hardness values for producing oral appliances [28]. Only the PJ-2 method showed a comparable surface hardness with the manual method, whereas the DLP and milling methods showed higher surface hardness than the manual method. Therefore, the surface hardness of the PJ-2, DLP, and milling methods was acceptable for replacing conventional acrylic resin.

Although the results of this study can be informative for clinicians who are using the conventional manual method but trying to adopt 3D printing technologies, careful clinical implementation is needed due to the following limitations: (1) a single resin material for each manufacturer and parameter was used in this study, even though types of resin and parameter settings, other than the 3D printing technology itself, are closely related to the physical and mechanical properties. Therefore, clinicians need to be cautious when using different resin materials and parameter settings. (2) Even though this in vitro study was conducted in a sophisticated experimental setting, actual oral environments are quite different. Oral environments are exposed to saliva and are subject to various changes in temperature, chemistry, and biology, which can affect the physical and mechanical properties of 3D-printed materials. Furthermore, moisture-driven dimensional change can affect the fit and function of the 3D-printed appliances. Therefore, further research including long-term dimensional accuracy in similar setting with oral environments or in vivo clinical trials would be necessary to obtain clinically meaningful results.

## 5. Conclusions

Within the limitations of this study,

- The most suitable 3D-printing method for manufacturing oral appliances was the DLP method, since this exhibited comparable characteristics with the manual method and met the ISO standards in terms of water absorption and solubility, flexural strength, and surface hardness.
- Alternatively, the FDM method can be used for manufacturing confined oral appliances, since this method exhibited comparable characteristics to the manual method and met the ISO standards in water absorption and solubility and flexural strength, despite the low surface hardness.
- The color stability of 3D-printing materials should be improved to manufacture oral appliances, which require long-term esthetics.

**Author Contributions:** Conceptualization, K.-G.H. and J.-S.A.; methodology, S.W.J., S.-H.P. and M.-K.L.; data curation, S.-W.L., H.-J.K. and M.-K.L.; writing—original draft preparation, H.-J.K. and S.-W.L.; writing—review and editing, J.-S.A. and K.-G.H.; visualization, H.-J.K. and S.-W.L.; project administration, K.-G.H. All authors have read and agreed to the published version of the manuscript.

**Funding:** This research was supported the National Research Foundation of Korea (NRF) funded by the Ministry of Education, Science and Technology (NRF-2017M3A9F1027928).

**Institutional Review Board Statement:** Not applicable.

**Informed Consent Statement:** Not applicable.

**Data Availability Statement:** The data that support the findings of this study are available upon reasonable request.

**Conflicts of Interest:** The authors declare no conflict of interest.

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
