# Peer review of "Which Three-Dimensional Printing Technology Can Replace Conventional Manual Method of Manufacturing Oral Appliance? A Preliminary Comparative Study of Physical and Mechanical Properties"

_applsci, doi:10.3390/app12010130_

Round 1

Reviewer 1 Report

The paper presents a comparative analysis of several 3D printing processes, and associated materials, for assessing their suitability for oral appliances. The comparison also includes milling as conventional manufacturing technology, as well as manual manufacturing. The criteria for the analysis were: color stability, flexural strength, hardness, solubility and absorption.

The paper results have practical implications; however several aspects should be detailed and clarified by the authors:

  • Present images of the types of devices for oral applications which can be manufactured by 3DP so that the readers to assess the complexity of the models and how important the dimensional accuracy of these models is. In this sense, please justify why dimensional accuracy after absorption tests was not assessed in this research. Did you notice dimensional changes after the absorption tests?
  • Please add the cost estimation for a type of printed oral device (at your choice) manufactured by each technology. This information is valuable as it offers a better perspective on replacing (or not) conventional manufacturing technology with 3DP technology. Especially because you have mentioned in conclusions that FDM is economic P11L334-335. If you bring into discussion an economic argument (which is a correct approach), then you have to present how you assess the cost-related aspects.
  • Please add a reference that can justify the statement in P2L58-59
  • Please justify why ABS-M30 was selected as material - to the best of my knowledge ABS M-30i is certified as a biocompatible material for 3DP. The same observation can be made for the other materials. Please document by relevant references if they are approved or not for medical/oral appliances.
  • What is the relevance of SEM investigations considering the analyzed criteria? Fractured surfaces of the analyzed specimens show particularities related to the manufacturing process and material, as expected. But how is this relevant for the purpose of the paper?
  • Please add supplementary material with information on the process parameters used for each manufacturing process. When stating that the surface hardness was lower for FDM than for the manual method, one can know exactly, for instance, what infill percentage was used and how many top/bottom layers were set when manufacturing FDM specimens. The assessed flexural properties or hardness, for all 3DP processes, are highly dependent on parameter settings. Other results are obtained when using other parameters settings. I consider that comment in this sense is mandatory in your study.

Reviewer 2 Report

Dear Authors,

I feel the topic of your paper is extremely interesting and of great interest to the readers of this journal, unfortunately before your article can be properly evaluated for publication you need to professionally proofread your article as many phrases are extremely badly written. I hope to receive your resubmission as soon as possible.

I have attached some comments below:

  1. You need to explain the acronyms in the tables.
  2. I believe the authors need to point out one major flaw of this article: for most 3d printing procedures only one resin of a single manufacturer was adopted, therefore their observations might be more related to the used product than to the manufacturing procedure.

Reviewer 3 Report

The paper “applsci-1481924-v1” was read.

I have some comments to improve the quality of the paper. My comments are listed below. Authors are encouraged to read and address them, then I can review the final version of the paper.

  1. Authors are encouraged to use the standard terminologies. Use ASTM 52900 for correct terminologies of AM processes. FFF is the commercial name of Stratasys and the correct terminology is Material Extrusion (MEX).
  2. The methodology of the paper needs to be highlighted at the end of the introduction.
  3. The contribution of the research needs to be highlighted.
  4. Additive manufacturing is a common and popular process to produce metal parts. To highlight the contribution of your paper I would suggest reading and adding the following paper to your introduction to show how additive manufacturing can be useful in different industries such as the aerospace industry. “Additive manufacturing a powerful tool for the aerospace industry.”
  5. Figure 2 shows nothing. Please remove it or provide a proper justification if you want to keep it in the text.
  6. Future work should be added to the paper.
  7. The introduction needs to be updated by the following new sources.
  • The potential of metal epoxy composite (MEC) as hybrid mold inserts in rapid tooling application: a review.
  • Additive manufacturing techniques for the fabrication of tissue engineering scaffolds: a review.
  • Design for additive manufacturing: a comprehensive review of the tendencies and limitations of methodologies.
  • Fused filament printing of specialized biomedical devices: a state-of-the art review of technological feasibilities with PEEK.
  • Numerical and analytical investigation on meltpool temperature of laser-based powder bed fusion of IN718

Round 2

Reviewer 1 Report

The comments from the previous review were answered by the authors. However, only a few modifications/additions were brought to the manuscript. Therefore, in my opinion, no significant improvements have been made. 

For instance, in the answer to my comment related to the necessity of SEM investigations, the authors mentioned the “surface micro-topography could give clinical implications in terms of the appliance management or hygienic control when the 3D printed appliance was fractured or cracked”. I don’t understand why this information was not included in the manuscript in the SEM corresponding section.

Where is figure 1 and its caption in the revised form of the manuscript? Are the occlusal splints in figure 1 manufactured for this study? Then why you did not test them along with the specimens?

I still believe that including in your comparative analysis materials which are not suitable for dental applications is not scientifically sound as it reduces the significance of the research and confuses the readers. It is hard to provide a logical argument for conducting research on materials you won’t use.

As the authors also acknowledge, the research has many limitations as being a preliminary one. Usually, in a high-ranked journal such as Applied Sciences, the complete research work is disseminated. Additional work is necessary.

Not even after the modifications, it is not clear for the reader what, actually, the purpose of the study is. It should have been clearly provided in the introductory section along with the research questions the study is answering and their relevance.

I am aware that there is much research investigating the effect of process parameters on the mechanical properties, but they are mostly performed on specimens. You also have functional objects - occlusal. That’s why I think providing transparent info on parameters is important.

Reviewer 2 Report

In the light of the substantive revision by the authors I think that the manuscript is suitable for publication.  

Author Response

Dear Reviewer,

We appreciated for accepting the manuscript.

The growth of the science and information base depends on the informed and thoughtful contributions of consultants like you.  
You encouraged our research, and advanced studies are expected to be performed in our team.

Kind regards,

Kyung-Gyun Hwang

Reviewer 3 Report

The paper is ready to publish. 

Author Response

(The authors gave the same response as above.)

Round 3

Reviewer 1 Report

The manuscript was improved. It is important that the title was modified to clearly include the fact that the research is its 'preliminary' phase.